# Brazil's Agribusiness Economic Miracle: Exploring Food Supply Chain Transformations for Promoting Win–Win Investments

José Elenilson Cruz [1,*], Gabriel da Silva Medina [2] and João Ricardo de Oliveira Júnior [3]

1. Federal Institute of Education, Science and Technology of Brasília, Brasília 70830-450, Brazil
2. Faculty of Agronomy and Veterinary Medicine, University of Brasilia (UnB), Brasilia 70910-900, Brazil; gabriel.silva.medina@gmail.com
3. Faculty of Management, Accounting and Economic Sciences, Universidade Federal de Goiás (UFG), Goiânia 74690-900, Brazil; joaoricjunior@gmail.com
* Correspondence: jose.cruz@ifb.edu.br

**Abstract:** *Background*: For many developing countries, agribusiness has become one of the main economic sectors, with the capacity to mobilize domestic and foreign investments. Despite the potential for development in countries like Brazil, the results of these investments in supply chains have not yet been systematically assessed. *Methods*: This study analyses foreign and domestic investments as an explanation for the recent growth of Brazilian agribusiness and evaluates the implications of different investment arrangements for the future development of the sector in the country. The research was based on a literature review of 12 agribusiness supply chains in Brazil. *Results*: Through a content analysis, the results reveal win–win situations with foreign and domestic investments supporting the streamlining of supply chains, mutually benefiting domestic and international groups and increasing the productivity of the entire sector. However, the results also reveal win–lose cases with chains and segments practically controlled by foreign multinationals in which local groups have practically no share. Finally, there are also cases of lose–win in which groups subsidized by the state are privileged in relation to others, compromising the sector's growth. *Conclusions*: The current liberal business environment results in the need for a new vision of development based on win–win opportunities for domestic and foreign investments created by dynamic sectors such as agribusiness.

**Keywords:** foreign direct investment (FDI); alternative food supply chain models; conceptualizations of food supply chain transformations; ongoing evolutions and transformations; patents

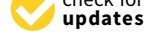



## 1. Introduction

For many low-income countries in South America, Africa, and Asia, the promotion of agribusiness is understood as an option from which to effectively benefit from global investment for the creation of urgently needed job opportunities and income from fees, taxes and exports, and to modernize and strengthen the domestic agricultural sector [1]. Agribusiness is the sum of all operations involved in the manufacture and distribution of farm supplies and production, storage, processing, and distribution of farm commodities [2].

Brazil's has become one of the prime examples of an economic boom promoted by growing investments in agribusiness in recent decades. In 2020, agribusiness as a whole (including supplies, industry, services, and agricultural production) accounted for 26.7% of Brazil's Gross Domestic Product (GDP), while agricultural production alone (primary sector of production in the field) accounted for about 7% of national GDP [3].

There are several reasons for the expansion of agribusiness in Brazil, such as land availability, favourable agrarian and environmental policies for the expansion of the agricultural frontier, agricultural policy's support for the modernization of rural producers through subsidized credit, and political support [4]. However, this favourable environment is not fulfilled without a fundamental aspect: investments. Investments play a fundamental role in explaining the economic miracle achieved by agribusiness in Brazil, and the knowledge

of the arrangements that favor ongoing investments is essential to envision the future of the sector.

Particularly since the 1990s, agribusiness has attracted considerable foreign direct investment (FDI), but it has also experienced significant private domestic investment and different public contributions in specific productive segments. The opening of Brazilian trade in the 1990s led to large investments in agribusiness in Brazil, mainly by foreign corporations, but the foreignization did not occur homogeneously in all supply chains or in all production segments. While some supply chains, such as soybean, began to rely on the predominance of foreign groups in their agro industrial sectors [5], other supply chains had more Brazilian investments, including investments in technology [6].

Therefore, understanding the arrangements that favor investments and their implications is fundamental in thinking about the future development of Brazilian agribusiness. The liberal and globalized business environment in which the country is inserted results in the need for a new outlook on development based on opportunities created by dynamic sectors such as agribusiness. A crucial challenge is the consolidation of domestic capital groups along the supply chains, overcoming the growing hegemony of foreign multinationals [7]. The agribusiness segments of the supply chains upstream and downstream of the farms tend to pay better than primary production on farms. This is because the industrial sector offers more opportunities for economies of scale than the agricultural sector, and the chaining and spillover effects are greater than in agriculture [8].

From the identification of the main market arrangements that have led to investments in agribusiness, this study aims to identify the origin of the predominant capital in the main sectors of agribusiness supply chains and analyze its implications for the future of the industry. We intend to analyze how ongoing investments leading to transformations in the agribusiness supply chains can offer strategic possibilities for growth in developing countries. Specifically, we intend to analyze: (1) the participation of foreign, domestic and public investments in the segments of important agribusiness supply chains in Brazil, and (2) the implications of these investments for the future of Brazilian agribusiness due to the possibilities created for domestic participation in win–win segments with better payoffs.

## 2. Theoretical Framework

Literature on foreign direct investment (FDI) by multinational enterprises has focused on outcomes for the host countries such as spillover effects, technology transfer, firm-level productivity, and performance of subsidiaries [9]. However, empirical evidence has also shown that the effects of FDI are heterogeneous and conditional on factors such as the type of FDI, the economic sector of investment, and the absorptive capacity of the host economy [10]. Productivity spillovers caused by FDI in Brazilian industry vary in terms of size, location, and the technological intensity of firms [11].

Existing studies, however, have only recently started exploring whether and to what degree domestic entrepreneurs can benefit from the economic dynamics promoted by FDI by establishing themselves in the marketplace while competing with multinational foreign enterprises [12]. Theoretically, liberal policies that encourage FDI may lead to: (1) business arrangements where domestic companies successfully compete with foreign companies and benefit from FDI or (2) business sectors controlled by foreign multinationals, with domestic groups having insignificant market shares and poorly benefiting from FDI [13]. In contrast, stronger governmental support may lead not only to significant domestic market shares but also to privileges and poor development.

Investment takes place when there is a direct interest of the parties concerned in a specific segment or economic sector [10]. Private direct investments are made by companies responding to market dynamics [9]. Public (governmental) investments are made through specific public programs and reforms [14]. In Brazil, specifically since the 1990s, the neoliberal economic perspective has been promoted through relaxed economic regulation and privatization policies [14]. With economic liberalization, the entry of international capital into the country boosted agribusiness and created a more competitive environment

for national groups [15]. However, a more sophisticated industrial base is a sine qua non condition for an emerging economy to converge from those already developed [16]. Therefore, it is necessary to create opportunities for domestic groups to increase their share in industrial sectors based on long-term policies, including industrial and technological policies [16].

The current situation of the liberal and globalized business environment in which the country operates results in the need for a clear assessment of the opportunities created by dynamic economic sectors such as agribusiness for domestic groups to thrive. A crucial challenge is the consolidation of companies with domestic capital throughout the supply chain of agribusiness in developing countries [17]. This challenge, certainly, ought to consider identifying the business arrangements most capable of absorbing the benefits of FDI, especially in terms of productivity, given the asymmetry in the levels of absorption of these benefits, as pointed out [11–18].

The role of investments can be involved in win–win, win–lose, and lose–lose arrangements [13–19]. Win–win outcomes occur when both sides benefit from the scenario; otherwise, win–lose situations result when only one side perceives the outcome as positive, and lose–lose means that all parties end up being worse off [19].

Building on this background, a key academic question that needs to be addressed is to what degree domestic entrepreneurs can establish themselves in the business and benefit from FDI which promotes dynamic economic sectors, such as is the case for agribusiness in Brazil in recent decades.

## 3. Methodology

In Brazil, measurements of the importance of agricultural production are made in Gross Value of Production, in accordance with the Brazilian Institute of Geography and Statistics (IBGE), and the relevance of agribusiness as a whole has been calculated in terms of GDP, as used by Cepea [3]. Finally, the relevance to the trade balance is estimated in currency. None of these measures, however, enables us to distinguish the extent of the participation of domestic groups in relation to foreign ones. Therefore, this study proposes the construction of an approach that considers participation in the market and the origin of the capital of the different companies acting in each segment.

To achieve the proposed objectives, this study was based on an integrative review [20,21] of empirical studies and on documental research carried out in institutional publications of sectoral organizations and companies. The integrative review of empirical studies followed the six steps proposed by Ercole et al. [21]. In the first stage, the research theme was delimited (participation of domestic capital in the agribusiness supply chains in Brazil). In the second stage, the criteria for inclusion and exclusion of studies were established, considering only empirical studies (articles and books) available in the Capes, Scielo and Google Academic databases. We selected 12 empirical articles published in scientific journals and 8 scientific studies published as book chapters.

In the third stage, the information to be extracted was defined at that related to the following keywords: "Brazilian participation", "Brazilian capital", "agribusiness", "production chain", in an interleaved manner and with the use of the Boolean operators "and" and "or", in Portuguese and English, in the title, abstract and keywords. At this stage, we prioritized studies that described aspects related to the following categories of analysis: (1) the main segments of the production chains in Brazil, (2) the activities developed by these segments, (3) the main companies operating in each productive segment (name, nationality and shareholding control), and (4) the market share of companies in the segments of the supply chains.

In order to standardize this information in all the analyzed production chains, it was necessary to carry out document research on institutional materials from sectoral associations and the companies themselves. To estimate the participation (market share) of the companies operating in each segment, first we quantified the total sales in the country for each input in each segment of the four supply chains (e.g., 5580 soybean harvesters sold

in Brazil in 2019), according to the assumptions established by Medina and Tomé [22]. We then identified the major international and domestic companies operating in each segment (e.g., CNH, John Deere, and AGCO in the case of soybean harvesters), and their total sales (e.g., 2903 soybean harvesters by CNH, 2269 by John Deere, and 408 by AGCO) [22]. To estimating the participation of domestic groups in relation to multinationals, we surveyed the shareholding composition of the companies as reported by Medina and Tomé [22]. To estimate the total market share of domestic groups in each segment of the production chain, the market shares of all companies with Brazilian capital were summed. The domestic participation in the production chain resulted from the weighted sum of the participation of business groups with Brazilian capital in each of the seven segments analysed (from seeds to marketing, see Figure 1).

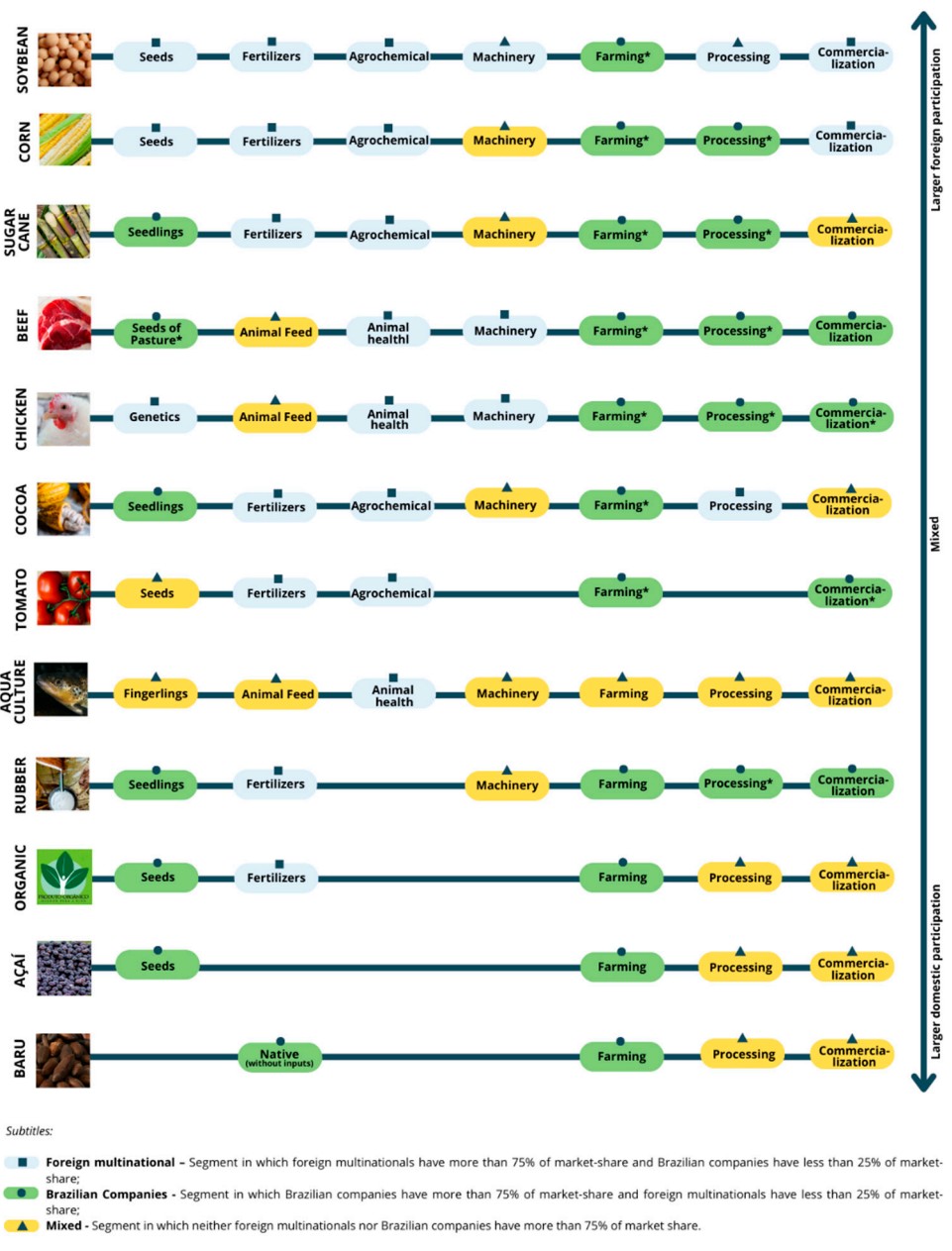

**Figure 1.** Participation of Brazilian and foreign economic groups in key segments of the supply chains analysed.

In the fourth stage, the aspects mentioned above were described for the 12 production chains studied. In the fifth stage, the results were interpreted, based on a discussion of the segments of the supply chains studied in which foreign capital predominates or domestic capital predominates (partly with state support), and the segments in which neither domestic nor international capital dominates. The sixth and final stage discusses the implications/understandings of the investment arrangements identified in the various segments of the supply chains.

Stages four, five, and six of the integrative review were conducted through the content analysis proposed by Bardin [23], being carried out in three phases: pre-analysis, material exploration, and treatment of the results [23]. In the pre-analysis, we carried out a preliminary review of the selected documents. In the second phase, we observed the themes that were repeated in the studies and chose the initial categories, i.e., the coding, classification, and categorization units [24]. Based on the content analysis, it was possible to group the initial categories and understand the recent growth of Brazilian agribusiness through three thematic categories: I. preponderance of foreign investments; II. the preponderance of investments made by domestic groups; III. mixed foreign and domestic investments without a clear preponderance. The third phase of the content analysis consisted of the treatment of the results through the inference and interpretation of the information collected in the integrative review.

## 4. Results

The supply chains analysed have different productive segments with distinct investment arrangements. Comparing the chains, trends were identified as follows: 1. There are cases of preponderance of foreign investments, as in the soybean and corn supply chains as a whole, and in segments associated with cutting edge technologies such as patented seeds, pesticides and animal health; 2. There are cases of preponderance of investments made by domestic groups, as in segments such as farming production and non-patented seeds; and 3. There are cases of segments with foreign and domestic investments without clear preponderance, which were called mixed segments (Figure 1). Throughout this results section, all the supply chains and segments evaluated in this study are presented, following the summary in Figure 1.

The set of chains presented has a total of 73 segments analysed. Of these segments, 25 (34%) are controlled mainly by foreign groups, 27 (37%) by domestic groups, and 21 (29%) are considered mixed without preponderant participation of domestic or foreign groups. Of the segments controlled by domestic groups, 12 (44.44%) are supported by direct public policies.

### 4.1. Soybean Supply Chain

The Brazilian market for transgenic seeds of soybean is firmly dominated by multinationals; specifically, the German company Bayer, with a market share of 90% [25]. Two-thirds of the profit from the final price of seeds remain in the hands of the multinational licensor, while the remaining 35% goes to seed producers, as they pay royalties for the use of patented transgenics [26]. In the segment of seed production, Brazilian companies hold 25% of the market share [27]. Thus, in the segment of seed production, domestic capital would be equivalent to only 8.7% (35% of the profits from the 25% market share) [27].

In the fertilizer segment, two types of companies operate, those that produce and those that use raw materials to manufacture specific fertilizer products. The multinational MOSAIC controls the raw material sector and the overall share of domestic groups has dropped to less than 9%. Concerning fertilizer manufacturers, the Brazilian market is dominated by the multinationals YARA and MOSAIC. Brazilian groups hold less than a third of the market, particularly the FERTIPAR Group and HERINGER. Brazilian participation in the fertilizer market can be estimated at less than 20% [27].

The agrochemical segment is divided into products with patents and generic products authorized after patent exclusivity periods. Product patents are fully controlled by

multinational groups. ChemChina (who bought SYNGENTA), BAYER, and BASF hold a significant market share. Generic products are very largely under the control of multinational companies, but some industries with domestic capital such as NORTOX and Ourofino Agrociência still have a stake. Overall, companies with national capital made less than 6% of the agrochemicals traded in Brazil [28].

The soybean-farming market for heavy machinery is controlled by a worldwide oligopoly characterized by mergers and acquisitions led by the following international groups: John Deere, CNH (holder of the brands Case and New Holland), and AGCO (holder of the brands Massey Ferguson and Valtra). The three groups combined control 99.6% of tractor sales and 100% of combine harvesters' sales in Brazil [29]. Agrale produces small-sized tractors with limited application in soybean farming and is the only relevant domestic company in this industry. There is a greater, but undefined, market share of domestic companies for agricultural implements such as ploughs, scarifiers, limestone spreaders, and cultivators.

Large multinational trading companies such as ADM, Bunge, Cargill, and Dreyfus (known as the ABCD Group) dominate the soybean processing and trading segments. Recently, China has massively invested in the segment of processing and trading, not only in Brazil but also in many other countries. In Brazil, the China National Cereals, Oils, and Foodstuffs Corporation (COFCO) purchased the Brazilian Noble Agri (trade). In total, domestic groups, including companies and farmers' cooperatives (e.g., Coamo and Comigo), control less than a fifth of the processing and trade of the soy produced in Brazil.

*4.2. Corn Supply Chain*

Corn is the basis for different supply chains such as pork, chicken, eggs and ethanol [30]. Corn production directly interferes with the chains involving products deriving from poultry, pork, milk and beef cattle, whereas the poultry and pork sector is highly dependent on this product [31].

The corn supply chain consists of the input sectors such as suppliers of pesticides, fertilizers, seeds, machinery and equipment; production itself (family or business producers); storage (cooperatives and public or private warehouses); processing (primary, covering the animal feed industry, the production of starch, corn flour and corn flakes; and secondary, including other end products, cereals, and cake mixes); distribution (for wholesale and retail, external and internal); consumption (from the farm to the chemical industry); institutional environment (legislation and government marketing mechanisms) and the organizational environment (bodies linked to technical assistance, credit and research) [32].

The Brazilian market is mostly dominated by multinational companies since it is one of the world leaders in the production of corn [33]. The Norwegian company Yara is the main owner of the occupation percentages within the fertilizer segment, and has a 4% Brazilian share [33]. The seed and agrochemical conglomerate is dominated by an oligopoly of the large companies Bayer, Syngenta and Corteva, justified by transgenic events that guarantee resistance to herbicides, insects or both [33].

As for machinery and implements, the Deere & Co group is responsible for more than 50% of the sector's revenues, with the Brazilian company Stara standing out, although with less than 1% of the market [33]. Finally, according to Corcioli et al. [33], marketing is the segment that moves the most resources within the corn production chain, with the highest revenues among the five segments, especially in the Cargill company, leader of the segment, and the Brazilian company Amaggi, which in 2019 had revenues of US$ 5 billion. Marketing is of paramount importance for producers; after all, it will lead to their financial results. Although part of the production is consumed in Brazil, some of it is exported. Thus, these companies have great relevance because they have the opportunity and the right conditions for large-scale acquisition to foster external demand [33].

There are two processes that give rise to industrialized corn products: dry milling (flours, snacks and breakfast cereals) and wet milling (oils, syrups and beverages) [34]. Approximately 70% of the corn produced in the world is destined for animal consumption [35].

The companies with the largest number of establishments authorized to manufacture feed in Brazil are: Seara Alimentos Ltd.a. (17.4% market share), BRF S.A. (market share of 12.5%), and Cargill Alimentos Ltd.a (market share of 8.9%) [33]. Regarding the participation of cooperatives in the production of animal feed, at least 300 sites were identified, especially the Aurora Alimentos Cooperative, with seven sites. Alfa Agro-industrial Cooperative and Catarinense Rural Agricultural Cooperative, both with six sites, C. Vale—Agro-industrial Cooperative, with five sites and Coamo Agro-industrial Cooperative, Copacol—Consolata Agro-industrial Cooperative and Lar Agro-industrial Cooperative, both with four sites [36].

*4.3. Sugarcane Supply Chain*

Contrary to soy, the breeding of sugarcane varieties is primarily a domestic domain, which largely reflects a significant promotion by public investment. Two-thirds of the sugarcane varieties cultivated in Brazil stem from the Inter-University Network for the Development of the Sugarcane Sector (RIDESA), a combination of ten universities. The other leading varieties are CTC, SP, IAC, and CV, representing 14%, 13%, 2%, 2%, and 4% of the planted area in Brazil, respectively [37].

The fertilizers and agrochemical market for sugarcane are similar to those for corn and soybean. Agricultural machinery for sugarcane farming includes harvesters, planters, sprayers, and trans-shipment trucks. The market for sugarcane harvesters is controlled by CNH and John Deere, which have by far the largest market shares [29]. In the case of planters, there is important participation by Brazilian groups such as DMB Máquinas e Implementos Agrícolas Ltd.a, TMA Máquinas (from the Tracan Group), and Sollus Agrícola. The Brazilian company Jacto, but also the French company Berthoud and multinationals AGCO (Valtra), CNH (Case), and John Deere, also operate and lead in the market for sprayers and other implements. Moreover, Brazilian groups mainly deliver sugarcane crushing industrial equipment. However, most of these groups act based on partnerships or joint ventures with multinational groups for the use, development, or import of technologies. Examples are Dedini S.A. Indústrias de Base, a domestic company that established a partnership with the Indian PRAJ industries in 2019, and Zanini Renk, a joint venture between the Brazilian Zanini and the German Renk AG for technology transfer from Germany to Brazil.

Regarding sugarcane mills, the situation is quite different. More than two-thirds of sugarcane processing is carried out in industrial plants held by Brazilian groups. In Brazil, there are 234 alcohol and sugar mills and another 178 alcohol distilleries. These 412 agro-industrial units process 643 million tons of sugarcane per year [38]. The Brazilian group Copersucar S.A. alone processes 85 million tons of sugar cane in 34 plants belonging to 20 different economic groups [39]. The Brazilian São Martinho Group leads the ranking for profitability [40]. Only recently, the segment has also attracted multinational groups. For example, the second-largest milling group is Raízen, a fifty–fifty joint venture between the Brazilian company Cosan S.A. and the multinational Royal Dutch Shell. BP British Petroleum formed a joint venture with Bunge within the newly created BP Bunge Bioenergia. The Atvos Agroindustrial group is moving from Brazilian controllers to American. Tereos Açúcar & Energia Brasil is part of the Tereos Internacional Group, a global French company. The Indian group Shree Renuka Sugars Ltd. Has also invested in the segment and today can process 13.6 million tons per year in Brazil.

Four large multinational groups control the Brazilian sugar market. However, Brazilian companies have created ethanol and sugar trading groups to increase their bargaining power vis-à-vis distributors [41]. Copersucar, for example, sells ethanol directly or through eco-energy, a trading company controlled by Copersucar. Sugar is sold through Alvean, a fifty–fifty joint venture formed by Copersucar and Cargill. The leader in the ethanol segment is the multinational Raízen, with 16.5 billion liters sold annually. Overall, domestic groups share 42.9% of the trade of sugar (23.1%) and ethanol (62.6%), totaling approximately 55.2% for the entire sugar segment.

## 4.4. Beef Supply Chain

The Brazilian market of pasture seeds is fragmented, but also sees a large participation from domestic groups. This reveals the lack of patented leading technology, which constitutes a barrier for market entry. Matsuda, a privately held Brazilian company, is a large player in this segment. The cultivars released by the Brazilian Agricultural Research Corporation Embrapa, mostly selected based on natural variability, account for more than 70% of the Brazilian forage seed market [42]. Recently, some multinationals have also started entering the market, for example, Barenbrug do Brasil, a company of the Royal Barenbrug Group based in the Netherlands, which started operating in Brazil in 2012.

The largest companies in the cattle feed segment are the multinationals Cargill and DSM. Together, they produce 15 million tons of feed per year, equivalent to 20% of the Brazilian market [43]. However, because of the high transport costs for heavy goods, the Brazilian feed market as a whole is in the hands of several small and large regional Brazilian companies. Among them, PREMIX stands out with a market share of 10%. Overall, the market share of domestic groups in the feed segment is estimated at 70.7% [42].

The animal health segment in Brazil is largely controlled by the four multinational groups: MSD, Zoetis, Boehringer Ingelheim, and Elanco, since they own the patents for all relevant state-of-the-art drugs [44]. MSD Saúde Animal is the veterinary arm of the American pharmaceutical Merck that bought the Brazilian veterinary industry Vallée in 2017, which was one of the leaders in the segment in the country. Zoetis, the actual leader in the global animal health market, was created after Pfizer Inc. decided to transform its animal health unit into an independent company. The largest group with domestic participation is the specialist in generic products Ourofino Saúde Animal, a publicly-traded company. Still, the original Brazilian shareholders hold 56.3% of the company. Another 16.9% is in the hands of the General Atlantic, a private equity company investing in growing companies. Other domestic companies are UCBVET, Calbos, Agener União, Real H, and JA.

The principal equipment used in beef cattle farming consists of containment trunks and weighing scales. A large number of domestic companies are active in this market segment because simple technologies require low initial investments and limited expertise [42]. Some companies such as Açôres have recently started investing in research to improve product performance and to search for alliances with multinational companies.

Officially, 67,058 cattle are slaughtered per day in Brazil [45]. The slaughterhouse segment is concentrated in three large public Brazilian companies: JBS, Marfrig, and Minerva [42]. JBS is a multinational controlled by the Brazilian company J & F Investimentos S.A. and has a broad range of shareholders: J & F Investimentos S.A. and Formosa with 39.8% share; a smaller treasury share (2.3%); BNDESPar, the investment branch of the Brazilian National Development Bank—BNDES (which also invested in Marfrig) with a 21.3% share; and other minor shareholders such as Brazilian public bank Caixa Econômica Federal (CEF) with 4.9% of the shares (JBS, 2020). JBS is the leading company in Brazil with an installed capacity to slaughter 34,200 heads of cattle per day, which corresponds to 51.0% of the Brazilian market. Likewise, JBS, Marfrig, and Minerva also went public, and domestic shareholding was estimated at 85% and 46.8%, respectively [46]. Despite market concentration in these three companies, there are another 1334 slaughterhouses registered by the federal inspection service [45].

## 4.5. Chicken Supply Chain

The poultry genetics segment in Brazil is controlled by two foreign multinationals: Aviagen and Cobb. The German group Erich Wesjohann (EW) controls Aviagen, and Cobb-Vantress, the poultry genetics arm of American Tyson Foods, is a world leader in the supply of poultry for broilers and in technical expertise in the poultry sector. Headquartered in Arkansas, United States, Cobb-Vantress has been present in Brazil for 22 years. By 2022, the company wants to reach the capacity to produce 42 million matrices, a number that includes the gaucho partner Agrogen [47]. In Brazil, only the two leaders in chicken meat (JBS/Seara and BRF) have the scale to buy poultry; the other industries buy matrices.

Considering the control of the multinationals, the participation of Brazilian groups was estimated at 1% in this market segment.

In the chicken feed segment, only animal nutrition companies market a portion of the feed, which corresponds to premixes and additives. The largest companies in the premixes and additives segment operating in Brazil are the multinationals Cargill and DSM. There are also Brazilian companies with a relevant share in the national animal nutrition market. The high cost of transportation, due to the weight of the products, ends up favoring regional groups. These factors, related to physical proximity and relationships, help explain why Brazilian groups hold 60.7% of the market [48].

Four multinational pharmaceutical groups control the animal health segment in Brazil: MSD, Zoetis, Boehringer Ingelheim and Elanco [49]. This control is largely related to the development and patenting of the latest technology drugs. Despite multinational control, domestic groups have an important share of the animal health market, particularly in the generic drug segment. Among the groups with domestic capital with significant market share, Ourofino and UCBVET stand out. The domestic share in the animal health segment in Brazil was estimated at 15.3% of the total.

The poultry chain has great demand for equipment. There are several categories of equipment, and the ten main categories are: slaughter, breeding, packaging, feed mill, freezing, hatchery, meat processing industrialization, laboratories, transportation and clothing. In this article, we considered only the breeding equipment that is acquired directly by the chicken producers from the commercial representatives of the manufacturing companies. This segment is mainly controlled by large multinational corporations, although there are competitive Brazilian companies with an estimated market share of 15% of the total market [48].

In the meatpacking segment, Brazilian multinationals JBS and BRF that control almost half of the market currently leads chicken meat production in Brazil. Other domestic groups with a tradition in Brazil control the rest of the market. In recent years, JBS has achieved leadership of the Brazilian broiler market by incorporating Céu Azul, Big Frango and Tyson. JBS is a multinational public listed company controlled by the Brazilian J & F Investimentos S.A. The participation of domestic groups in this segment of the chicken supply chain was estimated at 82.8%. This estimate was made considering only the Brazilian participation in the companies JBS and BRF (75% and 53.8% respectively) and the fact that all other companies in the segment are Brazilian [48].

*4.6. Cocoa Supply Chain*

Most of the 4.6 million tons of cocoa processed in 2020 occurred in Europe (36%), Oceania and Asia (24%), Africa (22%) and the Americas (19%) [50]. The largest continent (Africa) as a global producer of cocoa beans processed only one million tons, exporting the surplus, mainly to Europe, the continent that has the highest per capita consumption of chocolate in the world. In the Americas, the countries with the largest share in global cocoa processing are the United States (8%) and Brazil (5%) [51].

In Brazil, three multinational companies predominantly dominate the processing segment: Cargill, of American origin; Callebaut, from the Belgian group Barry-Callebaut; and Olam, of Nigerian origin, now controlled by Temasek Holdings (a Singaporean state company) and the Mitsubishi Corporation [52]. Together these companies account for 97% of national cocoa bean processing [53]. This concentration constitutes an oligopsony (i.e., few buyers) market structure [54]. Although most of the outputs of the cocoa processing link are directed to the domestic market and the smallest part to other countries, the trade balance with the latter is positive, unlike the situation in the processing link of other rural producers [54].

The insertion of Brazilian cocoa in the global market is basically restricted to the agricultural segment, which has structural shortcomings that compromise the competitiveness of the cocoa supply chain, and is predominantly represented by family farming, a segment that, although it plays a key role in ensuring food security in Brazil, traditionally faces

unfavorable competitive conditions compared to those for exporting agribusinesses [54]. The competitiveness of cocoa requires more favourable conditions for effective and sufficient access to resources capable of modifying the production structure of rural properties, technologies and technical support in order to ensure increased productivity [54].

On the other hand, the insertion in possibly more profitable arrangements, such as fine cocoa or vertical integration for the production of chocolates, also presents its own challenges, such as technological and knowledge barriers and increased transaction costs [54]. These barriers can, however, be mitigated with possible collective strategies aimed at producers, with the support of other organizations directly and indirectly interested in cocoa [54].

Whether via strategies to increase agricultural production or via insertion in potentially more profitable arrangements or even by combining both possibilities, these options do not concern only rural producers, but also the multiple organizations and actors directly or indirectly interested in the sector [54]. These strategies should be seen as a means of promoting the competitiveness of Brazilian cocoa in a context that favors social inclusion and the mitigation of its environmental impacts [54].

*4.7. Tomato Supply Chain*

In the tomato seed segment in Brazil, the companies with the largest market share are, respectively, Agristar, Syngenta AG, Monsoy, Blue Seeds and Sakata Seed [55]. Agristar, the market leader, is headquartered in the city of Santo Antônio de Posse, São Paulo, and has four experimental stations and a research and improvement unit in the states of São Paulo, Minas Gerais, Santa Catarina and Rio Grande do Norte. Syngenta AG, based in Basel (Switzerland), has been operating for 15 years with research and development activities focused on crop protection and seed production [56]. Monsoy, the current global vegetable seed branch of the German company Basf, operates in Brazil under the brand Nunhems. Blue Seeds, occupying the fourth position in the domestic market, is a national company based in Holambra/SP, with more than 20 years in the seed market aimed at the fruit and vegetable chain, covering the various soil and climate conditions in the country [56]. Sakata Seed, a Japanese company that produces and sells vegetable and flower seeds on the global market, entered Brazil in 1994 through the acquisition of Agroflora. It currently has more than 250 vegetable cultivars and 500 flower cultivars [56].

The agrochemicals segment in Brazil raised in 2019 the equivalent of US \$13.7 billion [56]. It is a concentrated market dominated by the companies Bayer CropScience, Syngenta, BASF, Corteva, FMC and UPL. Together, these companies control about 90% of the market [57].

The weakest and least coordinated link in the sector is in the production segment itself (inside the gate or on the farms) [56]. In the tomato chain, producers act in a more individualized and disarticulated way, sending their production to the State Supply Centres (wholesalers), selling directly to the retail sector or passing it on to middlemen, thus being at the mercy of unexpected changes in sale prices [56]. It is worth noting that, unlike industrial tomatoes, which experience a high degree of processing controlled by foreign multinationals, fresh tomatoes are marketed mainly by local agents [56].

It is worth mentioning that the production of tomatoes for fresh consumption mostly serves the domestic market, with the country participating with only 0.1% by weight of fresh or chilled tomato exports in the year 2017 [56]. The destination of the Brazilian product was the Mercosur countries, especially Argentina, while the main exporters were the states of Minas Gerais, São Paulo and Santa Catarina [58].

In recent years, in the tomato chain, the production of gourmet products and the creation of a brand associated with the product and its attributes have been a growing trend [56]. In this regard, the Trebeschi companies and the Mallmann group stand out on the national scene, maintaining their own production in protected fields and environments, for the most diverse gastronomic uses, with traditional and gourmet products that cater to different audiences [56].

*4.8. Aquaculture Supply Chain*

The world fish market is dominated by the following companies: Aquamaof (Revivim-Israel), Homey Group International (Shanghai-China), SalmonchilE (Santiago-Chile), Camanchaca (Santiago-Chile), Multiexport Foods (Puerto Montt-Chile), Cooke (Blacks Harbour-Canada), Rainforest (San Jose-Costa Rica), Regal Springs (Medan-Indonesia), Blue Gulf Seafoods (Shandong-China), Hainan Qinfu Industrial (Hainan-China), Expalsa (Guayas-Ecuador), Songa (Guayaquil-Ecuador) and Omarsa (Durán-Ecuador) [59]. These multinational companies, with a high degree of organization and production, can positively affect the Brazilian market, improving national productivity through new production technologies and genetic strains, but also negatively, taking international market shares from Brazilian companies or placing products with much higher level of competitiveness than Brazilian companies can achieve [59].

Data from IBGE [60] referring to the year 2019 show that the value of the production of young forms of fish is distributed in fish fry (65.97%), shrimp larvae and post-larvae (33.68%) and mollusk seeds (0.35%). The first are mainly composed of Nile tilapia fingerlings, whose main producers are the Aquagenetics Group (Aquabel and Aquamérica). Aquatec and Aquasul produce shrimp larvae and post-larvae, while shellfish seeds are distributed, almost exclusively, by UFSC [59].

In the feed manufacturing segment, the main players are Neovia, Guabi, Supra, Raguife and Comigo, for consumer fish, and Alcon, Nutricon, Maramar, Poytara, for ornamental fish (Rodrigues et al., 2021). With the exception of Raguife, Guabi and Comigo, which are Brazilian, the others have foreign capital participation [59].

For the production equipment segment, the domain is dominated by national industries, such as Alfakit, AcquaVita, Cardinal, Trevisan and Beraqua, regarding production equipment for broiler fish, and international companies, such as YSI and Horiba, regarding production equipment for ornamental fish [59]. Brazilian companies suffer strong competition from imported equipment with lower prices [59].

In the production segment (fattening), fish production in Brazil represents 88.39%, and shrimp and mollusk production 9.07% and 2.54%, respectively [59]. In terms of value (R$) of production, fish represent 73.26%, shrimp 25.14%, mollusks 1.47% and the other aquatic organisms only 0.13%. This primary production segment of the aquaculture supply chain represents in Brazil about 5 billion reais per year [59]. Although fish farming (pisciculture) in Brazil represents in volume the largest share of fish production, the average price per kilogram (kg) of shrimp is generally three times higher than the average kg values of the other two groups (fish and shellfish) [59]. The largest producers (fattening) of fish on the national scene are the foreign multinationals Ambar Amaral and Geneseas, and the Brazilian Copacol, C Vale and Tilabras (for Nilvo tilapia), Zaltana (for round fish) and NR Trutas (for trout). In shrimp production, Potyporã and Camanor stand out [59].

In the animal health segment, the main performers are Bayer and MSD, for international capital, and Aquivet, for national capital [59]. Danubio Piscicultura and Moana Aquacultura marketing the hormones used in hormone induction, and hormones for sexual reversion are imported from the foreign company FAV and distributed by the Brazilian company Nexco [59].

The processing and transformation segment of the aquaculture chain is a skilled industry with several operations, such as reception, gutting, washing, processing, packaging, freezing, storage, shipping and transport [59]. The main industrial plants are in the states of Paraná (Copacol, C. Vale and Brasilian Tilapia), São Paulo (Brasilian Fish, Mcassab), Mato Grosso do Sul (Geneseas), Mato Grosso (Delicious Fish), Minas Gerais (Coopeixe, Tilapia da Serra, NR Trout), Rondônia (Zaltana) and Goiás (Lake's Fish). The major companies use large imported processing equipment, and there is strong interest from foreign companies in investing in the segment and aiming at exporting to other countries [59].

*4.9. Rubber Supply Chain*

The rubber tree, which is native to the states of Amazonas, Acre, Pará, Roraima and Rondônia, began to be cultivated for economic purposes during the 1950s in several other Brazilian states [61]. Currently, extractivism still predominates in the northern region and cultivation (heveiculture) in the other states [61]. Despite being a country considered uncompetitive compared to the world's largest rubber producer (Thailand: 4.8 million tons), Brazil exported about 600 tons of rubber in 2018 to Latin American countries (FAO, 2021). Nevertheless, Brazil is a major importer of this product, as it imports over 60% of its consumption from countries such as Indonesia and Thailand [62]. This high level of importation makes the country often vulnerable to the international market, since the number of heveiculturers is still small, and the production system is based essentially on family farming, whose use of technology is low [63].

After being removed from the field, rubber is sent to 33 processing plants [62], most of which belong to the French company Michelin and the Brazilian companies Bráslatex, Hevea Tec, Colitex, QR Borrachas Quirino, Globorr, Noroeste Borracha, São Manuel, Agroindustrial ltuberá, Ask and SK [63]. Currently these plants are experiencing idle capacity and some are even economically unviable, due to the low latex supply, high demand and strong pressure from the automotive industry [64].

The main demanding party for processed rubber is the tire industry, as only 8% of this raw material is destined for other industries [65]. The main consumer companies for beneficiated rubber are Asian, European and North American tire manufacturers [63]. Goodyear, Michelin, Pirelli, Prometeon, Bridgestone, Continental and Sumitomo account for about 78% of the Market Share [63].

*4.10. Organic Supply Chain*

One of the major challenges of the organic supply chain in Brazil is the low availability of seeds [66]. The cultivation of organics in the national territory is dominated by conventional seeds [67], with some presence of imported seeds [66]. The pioneer companies of organic seeds in Brazil are Bionatur (RS), Isla (RS), Horticeres (MG) and Agristar, which launched its Naturalis line on the market, with seeds of 12 different vegetables [66]. In addition to these Brazilian companies, other major foreign players have entered the segment attracted by growth prospects, such as Koppert Biological Systems, Sumitomo Chemical, Bayer, Basf, Corteva and Syngenta [66].

The processing segment can be divided into two levels: primary processing and secondary processing [68]. Most of the companies that operate at the first level provide supplies and technical assistance—reproducing in part the integration process of other agro-food supply chains [66]—have their own brands, manage stands in supermarkets, and make sales directly to consumers and to secondary processing industries [66]. This group includes cooperatives or producer associations and companies with national capital [66]. Second-level companies generally use raw material from their own production [66], but also capture raw material from producers or primary processors [66]. This level includes several industries, ranging from traditional food industries, which use conventional production lines to process organics, to small cottage industries with specific production lines [69].

A movement of mergers and acquisitions has been observed in the organic product processing segment, as domestic industries have been bought by large corporations. Unilever acquired Mãe Terra in 2017, with the objectives of growing in the healthy products market in Brazil and internationalizing the brand [70]. The Paraná's Jasmine was bought by the French Nutrition et Sante in 2014, a company controlled by the Japanese Otsuka Nutraceuticals, a leader in the category of healthy, organic and functional products in Europe [66]. Thus, with the acquisition of Jasmine, Nutrition et Sante began to compete for leadership in the domestic market with the companies Nutrimental, Vital and Kobber [71].

### 4.11. Açaí Supply Chain

Four distinct systems predominate in the açaí supply chain: extractivism, management in floodplain areas, cultivation with irrigation, and cultivation without irrigation on the dry land areas [72]. Due to the high initial investment required, irrigated açaizeiro is recommended for medium and large-scale farmers. However, this does not rule out irrigated plantations by small farmers for those who can improvise irrigation systems with lower costs, taking advantage of watercourses or dams [73]. Pulp production, on the other hand, can be divided into two systems: artisanal (carried out by "beaters"), which supplies regional consumption, and large-scale (carried out by industrial processors) to supply the national market, especially the southeastern region, and the international market [74]. In the industrial processing segment (large-scale) the main companies are the American Sambazon and the Brazilian Cooperativa Agrícola Mista de Tome-Açu (CAMTA), Petruz Açaí, which exports to 35 countries in Europe, America, Asia and Africa, Bony Açaí, Palamaz, focused on the domestic market, Açaí Amazonas, focused on exports and the domestic market. These companies are able to meet, at national level, the specifications of distributors, usually limited to the content of total solids and sometimes pasteurization, and the international market, more rigorous in terms of food safety, sanitary conditions, pasteurization and complementary analysis (anthocyanin content, for example) and the laws of the destination countries. In addition to its use as food, açaí can be used in the cosmetics industry [75].

The flow of commercialization of açaí occurs on three levels. The first is defined by commercial transactions, between producers and buyers of the fruit at the production site, carried out under a perfect competition regime, except when the production is negotiated with agro-industries, in which a few buyers acquire a large part of the production of a given site. The second level is defined by commercial transactions between wholesalers, who gather a large volume of fruit, and local buyers. At this level, a small number of wholesalers set the resale price of the product for a large number of buyers [72]. The third level is defined by commercial transactions of açaí wine and derivatives in the retail market, where açaí greengrocers and churners operate under perfect competition, distributed in all neighborhoods of urban centres. At this level, the other products (blends, mix, pulp, ice cream, etc.) are also commercialized in supermarkets and special places, which have the power to set the selling price for consumers [72]. The domestication of the açaí is still in its initial steps and management practices still need further development in order to address environmental challenges and long-term maintenance [76].

### 4.12. Baru Supply Chain

The baru is a fruit native to the Brazilian Cerrado with production coming from nature, from collection by agro-extractivists on their properties, in common areas of agrarian reform settlements, and on large farms, with the authorization of the owners and upon payment of a charge on the amount collected [77]. The fruit is collected manually and only those fallen on the ground can be gathered [77]. The roasted chestnut is the most consumed and well-known product from the baru. However, there is research on the use of baru's pulp and peel. Rocha and Cardoso Santiago [78] developed wholegrain bread with pulp flour, which increased the nutrients in processed food. Other research has demonstrated the potential of the bark to be transformed into charcoal [79].

Stakeholders taking part in the baru production chain are: (i) agro-extractivists who collect the fruits and those who benefit from it, (ii) cooperatives, (iii) a network of intermediaries (companies, middlemen), and (iv) final consumer [77]. Baru has the specific dynamics of a native fruit, with all its production still coming from nature along with part of its artisanal processing [80]. Processing is currently done in two ways: artisanal and industrial. The former is performed by the agro-extractivists themselves, and the latter by the cooperative Copabase, with Brazilian capital, and by Barukas, a foreign company. The first uses only the nut of the fruit, roasting it and selling it on the national market to large industries, wholesalers and final consumers, at their own commercial points at fairs

and events. The second buys the whole fruit and exports it packaged in 50 kg bags [77], processes the roasting of the nut and extracts the fruit pulp.

In Brazil, Barukas offers the roasted nut in 90 g packages and the mix of the nut and the dehydrated baru pulp, also in 90 g packages [77]. In the United States, its portfolio of baru products consists of baru nuts with sea salt in 340 g packages, roasted baru nuts in 340 g packages, baru nuts with dark chocolate coating in 113 g packages, the mix of nuts and dehydrated baru pulp in 340 g packages, and baru butter in 227 g bottles [77].

## 5. Discussion

The comparative analysis of the different agribusiness supply chains based in Brazil makes it possible to explain the investments made in the sector and assess their implications for the future of the country's development. Based on the content analysis, it was possible to assess the recent growth of Brazilian agribusiness in three market arrangements: (I) preponderance of foreign investments; (II) preponderance of investments made by domestic groups; and (III) mixed foreign and domestic investments without a clear preponderance. On the one hand, there are supply chains and specific segments with greater participation and control by foreign multinational capital. On the other hand, there are specific segments with greater participation of domestic groups, in some cases with greater support from incentive policies. Finally, there are mixed segments in which domestic and foreign groups compete without consolidated control.

### 5.1. The Foreign Dominance of Part of Brazilian Agribusiness

Preponderance of foreign investments was observed both in supply chains as a whole and in specific segments of most of the chains studied. Illustrative examples include:

- Soybean and corn supply chains
- Transgenic seed segments with patents, high-tech machinery and state-of-the-art chemistry (not generics) including fertilizers, agrochemicals and animal health.

In most cases, these tend to be high technology sectors that require large investments and are often protected by patents. In these segments, the participation of domestic groups is smaller and often restricted to generic products. From this specific perspective, it can be said that patent protection in the country strengthened foreign control in some segments, such as transgenic seeds. Surprisingly, even in supply chains of organic products, there are segments with strong participation of foreign capital, as is the case with fertilizers.

In most cases, the segments that have attracted massive foreign investment are those in which domestic groups have failed to prosper. In the context of the country's agro-industrial development, these are segments that can be considered win–lose, given the advantages of foreign groups, which obtain most of the benefits, over domestic groups, which bear most of the associated risks and costs [11]. In these segments, the trend is the continued loss of domestic participation in the business, since the wealth generated today no longer contributes to the growth of local economic groups. This situation occurs in 25 (34%) of the 73 segments evaluated in the 12 supply chains researched.

### 5.2. Domestic Participation (Partly with State Support)

There are also supply chains and segments with greater control by domestic economic groups. In this case, agricultural production (primary production on farms) stands out, with practically all chains (except aquaculture) controlled by Brazilian producers. Examples of greater domestic participation also include:

- Baru, açaí, organic and natural rubber supply chains
- Farming production segments (rural producers), seeds for pasture in the beef supply chain, sugarcane varieties, and commercialization in several chains such as beef, chicken, tomatoes and rubber.

Characteristically, these are segments that rely on support from public policies, as in the case of farmers who benefit from subsidized credit from agricultural policy or the meat

processing segment (including JBS) that received contributions from the BNDESPar bank. In part, it can be said that these are segments subsidized by the state, generating lose–win situations that do not impact the increase in overall productivity [72], as they privilege only specific groups (farmers that access credit or companies directly supported by BNDESPar, for example) and not the entire sector from public money.

On the other hand, there are segments with more extensive state support, such as the development of pasture seeds and sugarcane varieties by research networks such as Embrapa and Ridesa in partnership with private domestic companies. These investments made in science and technology tend to support the development of the sector as a whole, not only specific groups, generating win–win situations that lead to increased productivity throughout the sector [19], more domestic public and private investments, and favor the opening of the market to foreign investments.

Of the 73 segments evaluated in the 12 supply chains studied, 27 (37%) are controlled by Brazilian groups. Of these, 12 (44%) received direct support from the state, of which two (17%) went to specific individuals and companies and 10 (83%) went to science and technology investments for the benefit of the sector in a diffuse manner.

*5.3. Mixed Segments—No Dominance*

We observed no dominance of investments by foreign corporations or domestic companies in the following segments and supply chains:

- With the exception of animal health, mixed segments dominate the other sectors in the aquaculture supply chain.
- Feed for cattle, chicken and fish, and machinery and processing and commercialization in several chains.

Typically, these are segments with medium-intensity technology, usually not protected by patents. These segments can be considered win–win because they rely on foreign and domestic private investments without the need for direct public investments. Of the 73 segments evaluated in the 12 supply chains studied, 21 (29%) are equivalent to mixed segments.

*5.4. Implications/Lessons on Investment Arrangements*

The economic opening of the 1990s boosted Brazilian agribusiness [15] by attracting foreign investments [12] that were added to domestic private and public investments [16]. Investments from different sources were identified in the supply chains analyzed:

(1) Foreign direct investments (FDI) leading to two types of arrangements. The first, of the win–lose type, promoted advantages for foreign groups in relation to domestic groups, and resulted in complete supply chains (soybean chain) and/or segments of several chains (patented transgenic seeds and high technology machinery and implements in the soybean and corn chains) dominated by foreign capital. The second, of the win–win type, promoted mutual gains for Brazilian and foreign companies from mixed segments in which domestic capital was combined with foreign capital for the development of certain sectors, such as feed for cattle and chicken, processing and marketing, and machinery and implements, in several of the chains studied.

(2) Private domestic investments of the win–win type, which promoted gains for national capital allowing the growth and consolidation of domestic groups in some segments, such as seedlings in the sugar cane, cocoa and rubber chains, seeds in the açaí and organic chains, animal feed in the beef and chicken chains, and commercialization in the beef and rubber chains.

(3) Public investments resulting in two arrangements. The first, of the win–win type, promoted the growth of domestic groups in some segments, such as pasture seed and sugarcane varieties, from the influx of investments in science and technology to the benefit of all. The second, of the lose–win type, promoted unequal growth among domestic groups by creating privileges for a few and benefiting specific production and processing (expansion of industrial plants) segments, with benefits for

some individuals (some large farmers) and organizations (beef processing giants for example).

This study adds to the existing literature on foreign direct investment (FDI) by multinational enterprises and the outcomes for the host countries [9]. It reveals whether and to what degree domestic entrepreneurs can benefit from the economic dynamics promoted by FDI by establishing themselves in the marketplace while competing with multinational foreign enterprises [12]. This study reveals that the effects of FDI are heterogeneous and conditional on local factors such as the absorptive capacity of the host economy [10]. We particularly reveal win–win scenarios in which both domestic and foreign investments supported dynamic segments of the agribusiness in Brazil.

## 6. Conclusions

Common sense has led Brazil to be seen as the prime example of agribusiness development worldwide. This study reveals that the reality is much more complex, with foreign multinational corporations controlling most of the agro-industrial segments of agribusiness carried out in Brazil and Brazilian companies having larger market shares mainly in the farming sector. Since agro-industrial segments can better remunerate capital and labor than farming, this study explores how domestic entrepreneurs can benefit from the thriving global agribusiness by establishing themselves in agro-industrial segments.

The ongoing investments and transformations in agribusiness supply chains offer new opportunities for economic growth in developing countries. Different market arrangements have provided for the allocation of large investments in Brazilian agribusiness, especially (1) foreign direct investments, (2) private domestic investments and (3) public investments in some segments. These different arrangements present in different supply chains are fundamental in explaining the recent great expansion of the sector in the country.

This study revealed that these different investments have distinct implications for the future of Brazilian agribusiness. Win–win arrangements rely on domestic investments and benefit from foreign investments, mutually benefiting domestic and international groups and increasing the productivity of the entire sector, and are thereby beneficial to the country in the short and long term. These are typical cases of the mixed segments in which there are no barriers preventing the entry of local groups (such as patents, very high technological knowledge or a great deal of invested capital). There are also some cases of policies aimed at developing technology in partnership with local companies. These are the arrangements that should be encouraged in the country.

Win–lose arrangements have led to the dominance of foreign groups in supply chains as a whole or in segments protected by patents or intensive in cutting-edge technology, hindering the entry of domestic capital. Lose–win arrangements have promoted unequal investments, based on economic subsidies that favor only a few to the detriment of many and do not contribute to the growth of the economic sector as a whole or even of the entire supply chain. These are the types of arrangements that should not be encouraged by Brazil, especially in the long run, since they result in more economic disadvantages for domestic groups.

**Author Contributions:** Conceptualization, J.E.C. and G.d.S.M.; methodology, J.E.C. and J.R.d.O.J.; validation, J.E.C., G.d.S.M. and J.R.d.O.J.; formal analysis, J.E.C., G.d.S.M. and J.R.d.O.J.; investigation, J.E.C. and G.d.S.M.; resources, J.E.C. and J.R.d.O.J.; writing—original draft preparation, J.E.C. and G.d.S.M.; writing—review and editing, J.E.C. and J.R.d.O.J.; All authors have read and agreed to the published version of the manuscript.

**Funding:** This research received no external funding.

**Data Availability Statement:** Not applicable.

**Conflicts of Interest:** The authors declare no conflict of interest.

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
