# Peer review of "Brazil’s Agribusiness Economic Miracle: Exploring Food Supply Chain Transformations for Promoting Win–Win Investments"

_logistics, 2022_

Round 1

Reviewer 1 Report

The reviewed article deals with the Brazil´s agribusiness and describes the specifics of individual selected commodities. The topic could be considered as an appropriate, suitable for wider research. The authors select as a key research question to what degree domestic entrepreneurs establish themselves as an agribusiness and benefit from FDI in Brazil.

Main purpose of the presented study is declared as a descriptive, seeks to describe the participation of domestic capital in agribusiness supply chain made in Brazil. While the main part of the article is only descriptive, there is missing wider methodological background. Current wider descriptive approach missing the analytical part, which further apply the findings from the description. There is also missing clear economic connection in the FDI, as it was highlighted at the initial part of the article. These two main points should be incorporated in the whole article idea and structure. Then it will be possible to apply appropriate methodological framework for further research, but not only descriptive. And, then it will be possible to identify clear findings as a research results and final conclusion, all in connection to FDI, as initially declared by authors.

Author Response

Dear Reviewer,

Thank you for your suggestions. We seek to meet them, improving the following aspects:

1) Methodology
2) Results and Discussion
3) Conclusion
4) References
5) Language of the text.

Sincerely,
The authors!

Reviewer 2 Report

The authors have presented an interesting approach in analyzing SCs and FDIs based on the literature. The paper is well-written and the methodology is quite well-supported. This means that there is room for improvement with respect to the presentation of the methodological choices and the corresponding limitations. I would finally like to suggest that the authors revisit the keywords they have chosen.

Author Response

(The authors gave the same response as above.)

Reviewer 3 Report

This paper may have much efforts covering formidable literature review.

As shown in authors' methodology, this paper finally suggests the interpretation about 12 agribusiness supply chains. Also, the various types of investment such as foreign dominated or dosmestic dominated or mixed are described in Figure 1.

Neverthless, authors mention win-lose type and win-win type in foreign direct investment and win-win type in private domestic investment and win-win type and lose-win type in public investment.

However, these private domestic investment and public investment were not distinguished in Chapter 4 and Chapter 5.1~5.3. Also, precise definitions of terms such as win-lose, win-win, and lose-win should be included respectively. Moreover, after these revisions each example for win-lose, win-win, and lose-win type according to degree of paticipation of foreign or private domestic or public domestic investment should be described in detail.

Author Response

(The authors gave the same response as above.)

Reviewer 4 Report

This study analyses foreign and domestic investments as an explanation for the recent growth of Brazilian agribusiness and evaluates the implications of different investment arrangements for the future development of the sector in the country. The research was based on a literature review of agribusiness supply chains in Brazil. The results reveal win-win situations with foreign and domestic investments supporting the streamlining of supply chains as in the sugarcane variety and pasture seed segments.

The subject of the paper is interesting. This paper contains new results and can be considered for publication after minor revision considering the following points: 

The abstract should be more prominent in which the authors should include important findings. 

Write down the future direction after the conclusion.

There are some mistakes in references like the year, vol, page number. Carefully check and corrected all references.

This paper should be edited grammatically.

Author Response

(The authors gave the same response as above.)

Round 2

Reviewer 1 Report

Based on the previous evaluation, the authors submitted a new version of the article, with incorporated comments. Despite the partial extension of the originally weaker methodological and analytical part, the overall concept of research is still limited primarily to the description. Instead of a deeper analysis through the application of some of the methods or tools of scientific work, the authors remain primarily on a descriptive level. For this reason, let me recommend to the authors to consider whether, based on the already performed evaluation of the current state of the researched issues, it would not be appropriate to choose a specific method of scientific work and apply it. This would increase the scientific level of the overall work already done.

Author Response

Dear Reviewer

In this round, we seek to better clarify the methodological techniques applied. Additionally, we inform that the methodological design chosen implied a descriptive Results Section, but we understand that our Discussion section has a more explanatory character.

Thanks for helping us improve the text.
The authors!

Reviewer 3 Report

Authors revised the paper properly following the comments.

However, there is just one thing that should be included.

As I commented in the first review, please define the private domestic investment and the public investment respectively in the proper section.

Author Response

Dear Reviewer

In response to your suggestion, we have included the definition of domestic public investment and private investment in the Theoretical framework section.

Thanks for helping us improve the text.
The authors!

Round 3

Reviewer 1 Report

Dear colleagues,

Thank you for your clarification and reaction on my review proposals. In fact, I could not find a revised version of your article in the review system, the corrected versions 1 and 2 seems to me identical.

In addition, in the current revised version, in the section where the results of your research are to be summarized, only other sources are cited. From the point of view of assessing the originality of the presented research, the question therefore arises, what part of the presented results are really your research and what is just a reference to the research presented by other authors in other articles. This is, for example, Chapter 4.11, in which almost all sentences are quoted from the source [68]. Similarly, Chapter 4.12, in which almost all sentences are quoted from the source [70].

For this reason, allow me to recommend adjusting your research concept to allow you to clearly present the results of your research, what has been found in your research, what follows, what are your research findings and recommendations.

Thank you for your understanding and cooperation.
